# Contour location via entropy reduction leveraging multiple information sources

**Alexandre N. Marques**
Department of Aeronautics and Astronautics
Massachusetts Institute of Technology
Cambridge, MA 02139
noll@mit.edu

**Remi R. Lam**
Center for Computational Engineering
Massachusetts Institute of Technology
Cambridge, MA 02139
rlam@mit.edu

**Karen E. Willcox**
Institute for Computational Engineering and Sciences
University of Texas at Austin
Austin, TX 78712
kwillcox@ices.utexas.edu

## Abstract

We introduce an algorithm to locate contours of functions that are expensive to evaluate. The problem of locating contours arises in many applications, including classification, constrained optimization, and performance analysis of mechanical and dynamical systems (reliability, probability of failure, stability, etc.). Our algorithm locates contours using information from multiple sources, which are available in the form of relatively inexpensive, biased, and possibly noisy approximations to the original function. Considering multiple information sources can lead to significant cost savings. We also introduce the concept of *contour entropy*, a formal measure of uncertainty about the location of the zero contour of a function approximated by a statistical surrogate model. Our algorithm locates contours efficiently by maximizing the reduction of contour entropy per unit cost.

## 1 Introduction

In this paper we address the problem of locating contours of functions that are expensive to evaluate. This problem arises in several areas of science and engineering. For instance, in classification problems the contour represents the boundary that divides objects of different classes. Another example is constrained optimization, where the contour separates feasible and infeasible designs. This problem also arises when analyzing the performance of mechanical and dynamical systems, where contours divide different behaviors such as stable/unstable, safe/fail, etc. In many of these applications, function evaluations involve costly computational simulations, or testing expensive physical samples. We consider the case when multiple information sources are available, in the form of relatively inexpensive, biased, and possibly noisy approximations to the original function. Our goal is to use information from all available sources to produce the best estimate of a contour under a fixed budget.

We address this problem by introducing the CLoVER (**C**ontour **Lo**cation **V**ia **E**ntropy **R**eduction) algorithm. CLoVER is based on a combination of principles from Bayesian multi-information source optimization [1–3] and information theory [4]. Our new contributions are:

- The concept of *contour entropy*, a measure of uncertainty about the location of the zero contour of a function approximated by a statistical surrogate model.

- An acquisition function that maximizes the reduction of contour entropy per unit cost.
- An algorithm that locates contours of functions using multiple information sources via reduction of contour entropy.

This work is related to the topic of Bayesian multi-information source optimization (MISO) [1–3, 5, 6]. Specifically, we use a statistical surrogate model to fit the available data and estimate the correlation between different information sources, and we choose the location for new evaluations as the maximizer of an acquisition function. However, we solve a different problem than Bayesian optimization algorithms. In the case of Bayesian optimization, the objective is to locate the global maximum of an expensive-to-evaluate function. In contrast, we are interested in the entire set of points that define a contour of the function. This difference is reflected in our definition of an acquisition function, which is fundamentally distinct from Bayesian optimization algorithms.

Other algorithms address the problem of locating the contour of expensive-to-evaluate functions, and are based on two main techniques: Support Vector Machine (SVM) and Gaussian process (GP) surrogate. CLoVER lies in the second category.

SVM [7] is a commonly adopted classification technique, and can be used to locate contours by defining the regions separated by them as different classes. Adaptive SVM [8–10] and active learning with SVM [11–13] improve the original SVM framework by adaptively selecting new samples in ways that produce better classifiers with a smaller number of observations. Consequently, these variations are well suited for situations involving expensive-to-evaluate functions. Furthermore, Dribusch et al. [14] propose an adaptive SVM construction that leverages multiple information sources, as long as there is a predefined fidelity hierarchy between the information sources.

Algorithms based on GP surrogates [15–20] use the uncertainty encoded in the surrogate to make informed decisions about new evaluations, reducing the overall number of function evaluations needed to locate contours. These algorithms differ mainly in the acquisition functions that are optimized to select new evaluations. Bichon et al. [15], Ranjan et al. [16], and Picheny et al. [17] define acquisition functions based on greedy reduction of heuristic measures of uncertainty about the location of the contour, whereas Bect et al. [18] and Chevalier et al. [19] define acquisition functions based on one-step look ahead reduction of quadratic loss functions of the probability of an excursion set. In addition, Stroh et al. [21] use a GP surrogate based on multiple information sources, under the assumption that there is a predefined fidelity hierarchy between the information sources. Opposite to the algorithms discussed above, Stroh et al. [21] do not use the surrogate to select samples. Instead, a pre-determined nested LHS design allocates the computational budget throughout the different information sources.

CLoVER has two fundamental distinctions with respect to the algorithms described above. First, the acquisition function used in CLoVER is based on one-step look ahead reduction of contour entropy, a formal measure of uncertainty about the location of the contour. Second, the multi-information source GP surrogate used in CLoVER does not require any hierarchy between the information sources. We show that CLoVER outperforms the algorithms of Refs. [15–20] when applied to two problems involving a single information source. One of these problems is discussed in Sect. 4, while the other is discussed in the supplementary material.

The remainder of this paper is organized as follows. In Sect. 2 we present a formal problem statement and introduce notation. Then, in Sect. 3 we introduce the details of the CLoVER algorithm, including the definition of the concept of contour entropy. Finally, in Sect. 4 we present examples that illustrate the performance of CLoVER.

## 2  Problem statement and notation[1]

Let $g : \mathcal{D} \mapsto \mathbb{R}$ denote a continuous function on the compact set $\mathcal{D} \in \mathbb{R}^d$, and $g_\ell : \mathcal{D} \mapsto \mathbb{R}$, $\ell \in [M]$, denote a collection of the $M$ information sources (IS) that provide possibly biased estimates of $g$. (For $M \in \mathbb{Z}^+$, we use the notation $[M] = \{1, \ldots, M\}$ and $[M]_0 = \{0, 1, \ldots, M\}$). In general, we assume that observations of $g_\ell$ may be noisy, such that they correspond to samples from the normal distribution $\mathcal{N}(g_\ell(\boldsymbol{x}), \lambda_\ell(\boldsymbol{x}))$. We further assume that, for each IS $\ell$, the query cost function,

$c_\ell : \mathcal{D} \mapsto \mathbb{R}^+$, and the variance function $\lambda_\ell$ are known and continuously differentiable over $\mathcal{D}$. Finally, we assume that $g$ can also be observed directly without bias (but possibly with noise), and refer to it as information source 0 (IS0), with query cost $c_0$ and variance $\lambda_0$.

Our goal is to find the best approximation, within a fixed budget, to a specific contour of $g$ by using a combination of observations of $g_\ell$. In the remainder of this paper we assume, without loss of generality, that we are interested in locating the zero contour of $g$, defined as the set $\mathcal{Z} = \{z \in \mathcal{D} \mid g(z) = 0\}$.

## 3 The CLoVER algorithm

In this section we present the details of the CLoVER (**C**ontour **Lo**cation **V**ia **E**ntropy **R**eduction) algorithm. CLoVER has three main components: (i) a statistical surrogate model that combines information from all $M+1$ information sources, presented in Sect. 3.1, (ii) a measure of the entropy associated with the zero contour of $g$ that can be computed from the surrogate, presented in Sect. 3.2, and (iii) an acquisition function that allows selecting evaluations that reduce this entropy measure, presented in Sect. 3.3. In Sect. 3.4 we discuss the estimation of the hyperparameters of the surrogate model, and in Sect. 3.5 we show how these components are combined to form an algorithm to locate the zero contour of $g$. We discuss the computational cost of CLoVER in the supplementary material. An implementation of CLoVER in Python 2.7 is available at https://github.com/anmarques/CLoVER.

### 3.1 Statistical surrogate model

CLoVER uses the statistical surrogate model introduced by Poloczek et al. [3] in the context of multi-information source optimization. This model constructs a single Gaussian process (GP) surrogate that approximates all information sources $g_\ell$ simultaneously, encoding the correlations between them. Using a GP surrogate allows data assimilation using standard tools of Gaussian process regression [22].

We denote the surrogate model by $f$, with $f(\ell, \boldsymbol{x})$ being the normal distribution that represents the belief about IS $\ell$, $\ell \in [M]_0$, at location $\boldsymbol{x}$. The construction of the surrogate follows from two modeling choices: (i) a GP approximation to $g$ denoted by $f(0, \boldsymbol{x})$, i.e., $f(0, \boldsymbol{x}) \sim GP(\mu_0, \Sigma_0)$, and (ii) independent GP approximations to the biases $\delta_\ell(\boldsymbol{x}) = g_\ell(\boldsymbol{x}) - g(x)$, $\delta_\ell \sim GP(\mu_\ell, \Sigma_\ell)$. Similarly to [3], we assume that $\mu_\ell$ and $\Sigma_\ell$, $\ell \in [M]_0$, belong to one of the standard parameterized classes of mean functions and covariance kernels. Finally, we construct the surrogate of $g_\ell$ as $f(\ell, \boldsymbol{x}) = f(0, \boldsymbol{x}) + \delta_\ell(\boldsymbol{x})$. As a consequence, the surrogate model $f$ is a GP, $f \sim GP(\mu, \Sigma)$, with

$$\mu(\ell, \boldsymbol{x}) = \mathbb{E}[f(\ell, \boldsymbol{x})] = \mu_0(\boldsymbol{x}) + \mu_\ell(\boldsymbol{x}), \tag{1}$$

$$\Sigma\big((\ell, \boldsymbol{x}), (m, \boldsymbol{x}')\big) = \mathrm{Cov}\big(f(\ell, \boldsymbol{x}), f(m, \boldsymbol{x}')\big) = \Sigma_0(\boldsymbol{x}, \boldsymbol{x}') + \mathbb{1}_{\ell,m}\Sigma_\ell(\boldsymbol{x}, \boldsymbol{x}'), \tag{2}$$

where $\mathbb{1}_{\ell,m}$ denotes the Kronecker's delta.

### 3.2 Contour entropy

In information theory [4], the concept of entropy is a measure of the uncertainty in the outcome of a random process. In the case of a discrete random variable $W$ with $k$ distinct possible values $w_i$, $i \in [k]$, entropy is defined by

$$H(W) = -\sum_{i=1}^{k} P(w_i) \ln P(w_i), \tag{3}$$

where $P(w_i)$ denotes the probability mass of value $w_i$. It follows from this definition that lower values of entropy are associated to processes with little uncertainty ($P(w_i) \approx 1$ for one of the possible outcomes).

We introduce the concept of *contour entropy* as the entropy of a discrete random variable associated with the uncertainty about the location of the zero contour of $g$, as follows. For any given $\boldsymbol{x} \in \mathcal{D}$, the posterior distribution of $f(0, \boldsymbol{x})$ (surrogate model of $g(\boldsymbol{x})$), conditioned on all the available evaluations, is a normal random variable with known mean $\mu(0, \boldsymbol{x})$ and variance $\sigma^2(0, \boldsymbol{x})$. Given $\epsilon(\boldsymbol{x}) \in \mathbb{R}^+$, an observation $y$ of this random variable can be classified as one of the following three

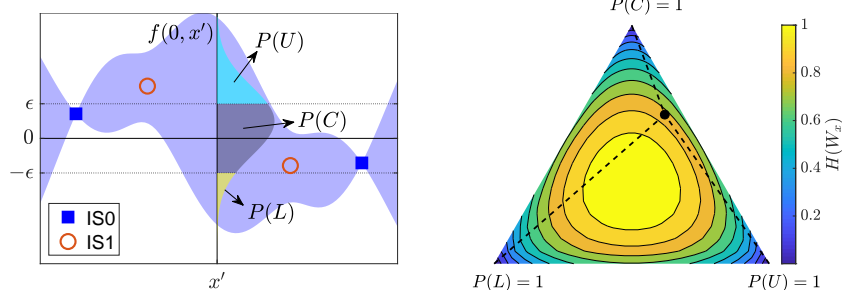

Figure 1: Left: GP surrogate, distribution $f(0, x')$ and probability mass of events $L$, $C$, and $U$, which define the random variable $W_{x'}$. Right: Entropy $H(W_x)$ as a function of the probability masses. The black dot corresponds to $H(W_{x'})$.

events: $y < -\epsilon(\boldsymbol{x})$ (denoted as event $L$), $|y| < \epsilon(\boldsymbol{x})$ (denoted as event $C$), or $y > \epsilon(\boldsymbol{x})$ (denoted as event $U$). These three events define a discrete random variable, $W_{\boldsymbol{x}}$, with probability mass $P(L) = \Phi((-\mu(0, \boldsymbol{x}) - \epsilon(\boldsymbol{x}))/\sigma(0, \boldsymbol{x}))$, $P(C) = \Phi((-\mu(0, \boldsymbol{x}) + \epsilon(\boldsymbol{x}))/\sigma(0, \boldsymbol{x})) - \Phi((-\mu(0, \boldsymbol{x}) - \epsilon(\boldsymbol{x}))/\sigma(0, \boldsymbol{x}))$, $P(U) = \Phi((\mu(0, \boldsymbol{x}) - \epsilon(\boldsymbol{x}))/\sigma(0, \boldsymbol{x}))$, where $\Phi$ is the unit normal cumulative distribution function. Figure 1 illustrates events $L$, $C$, and $U$, and the probability mass associated with each of them. In particular, $P(C)$ measures the probability of $g(\boldsymbol{x})$ being within a band of width $2\epsilon(\boldsymbol{x})$ surrounding the zero contour, as estimated by the GP surrogate. The parameter $\epsilon(\boldsymbol{x})$ represents a tolerance in our definition of a zero contour. As the algorithm gains confidence in its predictions, it is natural to reduce $\epsilon(\boldsymbol{x})$ to tighten the bounds on the location of the zero contour. As discussed in the supplementary material, numerical experiments indicate that $\epsilon(\boldsymbol{x}) = 2\sigma(\boldsymbol{x})$ results in a good balance between exploration and exploitation.

The entropy of $W_{\boldsymbol{x}}$ measures the uncertainty in whether $g(\boldsymbol{x})$ lies below, within, or above the tolerance $\epsilon(\boldsymbol{x})$, and is given by

$$H(W_{\boldsymbol{x}}; f) = -P(L) \log P(L) - P(C) \log P(C) - P(U) \log P(U). \tag{4}$$

This entropy measures uncertainty at parameter value $\boldsymbol{x}$ only. To characterize the uncertainty of the location of the zero contour, we define the *contour entropy* as

$$\mathscr{H}(f) = \frac{1}{V(\mathcal{D})} \int_{\mathcal{D}} H(W_{\boldsymbol{x}}; f) \, d\boldsymbol{x}, \tag{5}$$

where $V(\mathcal{D})$ denotes the volume of $\mathcal{D}$.

### 3.3 Acquisition function

CLoVER locates the zero contour by selecting samples that are likely reduce the contour entropy at each new iteration. In general, samples from IS0 are the most informative about the zero contour of $g$, and thus are more likely to reduce the contour entropy, but they are also the most expensive to evaluate. Hence, to take advantage of the other $M$ IS available, the algorithm performs observations that maximize the expected reduction in contour entropy, normalized by the query cost.

Consider the algorithm after $n$ samples evaluated at $X_n = \{(\ell^i, \boldsymbol{x}^i)\}_{i=1}^n$, which result in observations $Y_n = \{y^i\}_{i=1}^n$. We denote the posterior GP of $f$, conditioned on $\{X_n, Y_n\}$, as $f^n$, with mean $\mu^n$ and covariance matrix $\Sigma^n$. Then, the algorithm selects a new parameter value $\boldsymbol{x} \in \mathcal{D}$, and IS $\ell \in [M]_0$ that satisfy the following optimization problem.

$$\underset{\ell \in [M]_0, \, \boldsymbol{x} \in \mathcal{D}}{\text{maximize}} \quad u(\ell, \boldsymbol{x}; f^n), \tag{6}$$

where

$$u(\ell, \boldsymbol{x}; f^n) = \frac{\mathbb{E}_y[\mathscr{H}(f^n) - \mathscr{H}(f^{n+1}) \mid \ell^{n+1} = \ell, \, \boldsymbol{x}^{n+1} = \boldsymbol{x}]}{c_\ell(\boldsymbol{x})}, \tag{7}$$

and the expectation is taken over the distribution of possible observations, $y^{n+1} \sim \mathcal{N}\big(\mu^n(\ell, \boldsymbol{x}), \Sigma^n((\ell, \boldsymbol{x}), (\ell, \boldsymbol{x}))\big)$. To make the optimization problem tractable, the

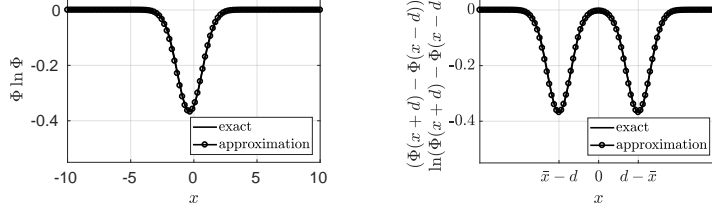

Figure 2: Comparison between functions involving products of $\Phi$ and $\ln \Phi$ and approximations (8–9).

search domain is replaced by a discrete set of points $\mathcal{A} \subset \mathcal{D}$, e.g., a Latin Hypercube design. We discuss how to evaluate the acquisition function $u$ next.

Given that $f^n$ is known, $\mathscr{H}(f^n)$ is a deterministic quantity that can be evaluated from (4–5). Namely, $H(W_x; f^n)$ follows directly from (4), and the integration over $\mathcal{D}$ is computed via a Monte Carlo-based approach (or regular quadrature if the dimension of $\mathcal{D}$ is relatively small).

Evaluating $\mathbb{E}_y[\mathscr{H}(f^{n+1})]$ requires a few additional steps. First, the expectation operator commutes with the integration over $\mathcal{D}$. Second, for any $\boldsymbol{x}' \in \mathcal{D}$, the entropy $H(W_{\boldsymbol{x}'}; f^{n+1})$ depends on $y^{n+1}$ through its effect on the mean $\mu^{n+1}(0, \boldsymbol{x}')$ (the covariance matrix $\Sigma^{n+1}$ depends only on the location of the samples). The mean is affine with respect to the observation $y^{n+1}$ and thus is distributed normally: $\mu^{n+1}(0, \boldsymbol{x}') \sim \mathcal{N}(\mu^n(0, \boldsymbol{x}'), \bar{\sigma}^2(\boldsymbol{x}'; \ell, \boldsymbol{x}))$, where $\bar{\sigma}^2(\boldsymbol{x}'; \ell, \boldsymbol{x}) = \left(\Sigma^n((0, \boldsymbol{x}'), (\ell, \boldsymbol{x}))\right)^2 / \left(\lambda_\ell(\boldsymbol{x}) + \Sigma^n((\ell, \boldsymbol{x}), (\ell, \boldsymbol{x}))\right)$. Hence, after commuting with the integration over $\mathcal{D}$, the expectation with respect to the distribution of $y^{n+1}$ can be equivalently replaced by the expectation with respect to the distribution of $\mu^{n+1}(0, \boldsymbol{x}')$, denoted by $\mathbb{E}_\mu[(.)]$.

Third, in order to compute the expectation operator analytically, we introduce the following approximations.

$$\Phi(x) \ln \Phi(x) \approx \sqrt{2\pi}\, c\varphi(x - \bar{x}), \tag{8}$$

$$(\Phi(x + d) - \Phi(x - d)) \ln(\Phi(x + d) - \Phi(x - d)) \approx \sqrt{2\pi}\, c\big(\varphi(x - d + \bar{x}) + \varphi(x + d - \bar{x})\big), \tag{9}$$

where $\varphi$ is the normal probability density function, $\bar{x} = \Phi^{-1}(e^{-1})$, and $c = \Phi(\bar{x}) \ln \Phi(\bar{x})$. Figure 2 shows the quality of these approximations. Then, we can finally write

$$\mathbb{E}_y[\mathscr{H}(f^{n+1}) \mid \ell^{n+1} = \ell, \boldsymbol{x}^{n+1} = \boldsymbol{x}]$$

$$= \frac{1}{V(\mathcal{D})} \int_{\mathcal{D}} \mathbb{E}_\mu[H(W_{\boldsymbol{x}'}; f^{n+1}) | \ell^{n+1} = \ell, \boldsymbol{x}^{n+1} = \boldsymbol{x}] \, d\boldsymbol{x}'$$

$$\approx -\frac{c}{V(\mathcal{D})} \int_{\mathcal{D}} r_\sigma(\boldsymbol{x}; \ell, \boldsymbol{x}) \sum_{i=0}^{1} \sum_{j=0}^{1} \exp\left(-\frac{1}{2}\left(\frac{\mu^n(0, \boldsymbol{x}') + (-1)^i \epsilon}{\hat{\sigma}(\boldsymbol{x}'; \ell, \boldsymbol{x})} + (-1)^j \bar{x} r_\sigma(\boldsymbol{x}'; \ell, \boldsymbol{x})\right)^2\right) d\boldsymbol{x}', \tag{10}$$

where

$$\hat{\sigma}^2(\boldsymbol{x}'; \ell, \boldsymbol{x}) = \Sigma^{n+1}((0, \boldsymbol{x}'), (0, \boldsymbol{x}')) + \bar{\sigma}^2(\boldsymbol{x}'; \ell, \boldsymbol{x}), \quad r_\sigma^2(\boldsymbol{x}'; \ell, \boldsymbol{x}) = \frac{\Sigma^{n+1}((0, \boldsymbol{x}'), (0, \boldsymbol{x}'))}{\hat{\sigma}^2(\boldsymbol{x}'; \ell, \boldsymbol{x})}.$$

## 3.4 Estimating hyperparameters

Our experience indicates that the most suitable approach to estimate the hyperparameters depends on the problem. Maximum a posteriori (MAP) estimates normally perform well if reasonable guesses are available for the priors of hyperparameters. On the other hand, maximum likelihood estimates (MLE) may be sensitive to the randomness of the initial data, and normally require a larger number of evaluations to yield appropriate results.

Given the challenge of estimating hyperparameters with small amounts of data, we recommend updating these estimates throughout the evolution of the algorithm. We adopt the strategy of

estimating the hyperparameters whenever the algorithm makes a new evaluation of IS0. The data obtained by evaluating IS0 is used directly to estimate the hyperparameters of $\mu_0$ and $\Sigma_0$. To estimate the hyperparameters of $\mu_\ell$ and $\Sigma_\ell$, $\ell \in [M]$, we evaluate all other $M$ information sources at the same location and compute the biases $\delta_\ell = y_\ell - y_0$, where $y_\ell$ denotes data obtained by evaluating IS $\ell$. The biases are then used to estimate the hyperparameters of $\mu_\ell$ and $\Sigma_\ell$.

### 3.5 Summary of algorithm

1. Compute an initial set of samples by evaluating all $M + 1$ IS at the same values of $x \in \mathcal{D}$. Use samples to compute hyperparameters and the posterior of $f$.

2. Prescribe a set of points $\mathcal{A} \subset \mathcal{D}$ which will be used as possible candidates for sampling.

3. Until budget is exhausted, do:

   (a) Determine the next sample by solving the optimization problem (6).
   (b) Evaluate the next sample at location $x^{n+1}$ using IS $\ell^{n+1}$.
   (c) Update hyperparameters and posterior of $f$.

4. Return the zero contour of $\mathbb{E}[f(0, x)]$.

## 4 Numerical results

In this section we present three examples that demonstrate the performance of CLoVER. The first two examples involve multiple information sources, and illustrate the reduction in computational cost that can be achieved by combining information from multiple sources in a principled way. The last example compares the performance of CLoVER to that of competing GP-based algorithms, showing that CLoVER can outperform existing alternatives even in the case of a single information source.

### 4.1 Multimodal function

In this example we locate the zero contour of the following function within the domain $\mathcal{D} = [-4, 7] \times [-3, 8]$.

$$g(x) = \frac{(x_1^2 + 4)(x_2 - 1)}{20} - \sin\left(\frac{5x_1}{2}\right) - 2. \tag{11}$$

This example was introduced in Ref. [15] in the context of reliability analysis, where the zero contour represents a failure boundary. We explore this example further in the supplementary material, where we compare CLoVER to competing algorithms in the case of a single information source. To demonstrate the performance of CLoVER in the presence of multiple information sources, we introduce the following biased estimates of $g$:

$$g_1(x) = g(x) + \sin\left(\frac{5}{22}\left(x_1 + \frac{x_2}{2}\right) + \frac{5}{4}\right), \quad g_2(x) = g(x) + 3\sin\left(\frac{5}{11}(x_1 + x_2 + 7)\right).$$

We assume that the query cost of each information source is constant: $c_0 = 1$, $c_1 = 0.01$, $c_2 = 0.001$. We further assume that all information sources can be observed without noise.

Figure 3 shows predictions made by CLoVER at several iterations of the algorithm. CLoVER starts with evaluations of all three IS at the same 10 random locations. These evaluations are used to compute the hyperparameters using MLE, and to construct the surrogate model. The surrogate model is based on zero mean functions and squared exponential covariance kernels [22]. The contour entropy of the initial setup is $\mathcal{H} = 0.315$, which indicates that there is considerable uncertainty in the estimate of the zero contour. CLoVER proceeds by exploring the parameter space using mostly IS2, which is the model with the lowest query cost. The algorithm stops after 123 iterations, achieving a contour entropy of $\mathcal{H} = 4 \times 10^{-9}$. Considering the samples used in the initial setup, CLoVER makes a total of 17 evaluations of IS0, 68 evaluations of IS1, and 68 evaluations of IS2. The total query cost is 17.8. We repeat the calculations 100 times using different values for the initial 10 random evaluations, and the median query cost is 18.1. In contrast, the median query cost using a single information source (IS0) is 38.0, as shown in the supplementary material. Furthermore, at query cost 18.0, the median contour entropy using a single information source is $\mathcal{H} = 0.19$.

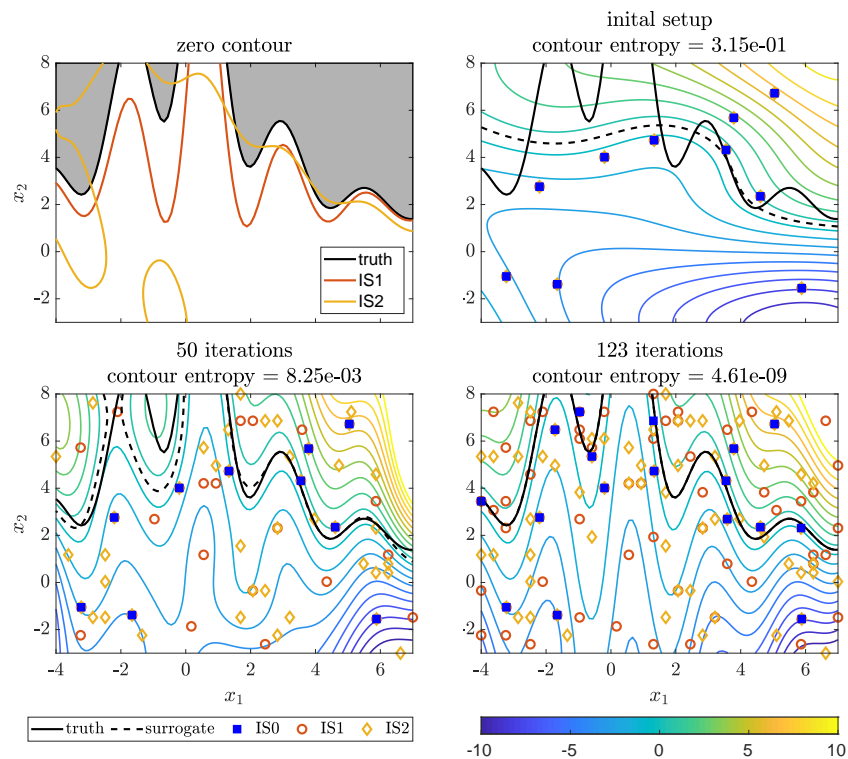

Figure 3: Locating the zero contour of the multimodal function (11). Upper left: Zero contour of IS0, IS1, and IS2. Other frames: Samples and predictions made by CLoVER at several iterations. Dashed black line: Zero contour predicted by the surrogate model. Colors: Mean of the surrogate model $f(0, \boldsymbol{x})$. CLoVER obtains a good approximation to the zero contour with only 17 evaluations of expensive IS0.

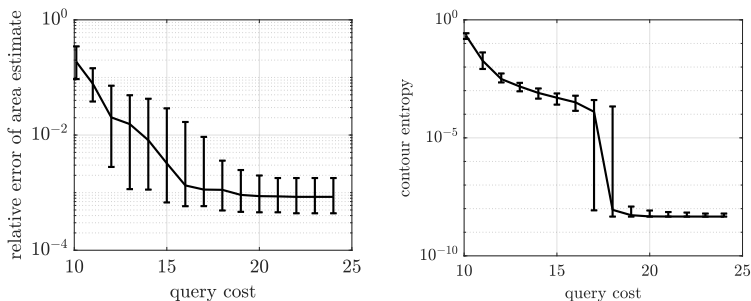

Figure 4: Left: Relative error in the estimate of the area of the set $S$. Right: Contour entropy. Median, 25, and 75 percentiles.

We assess the accuracy of the zero contour estimate produced by CLoVER by measuring the area of the set $S = \{ \boldsymbol{x} \in \mathcal{D} \mid g(\boldsymbol{x}) > 0 \}$ (shaded region shown on the top left frame of Figure 3). We estimate the area using Monte Carlo integration with $10^6$ samples in the region $[-4, 7] \times [1.4, 8]$. We compute a reference value by averaging 20 Monte Carlo estimates based on evaluations of $g$: area$(S) = 36.5541$. Figure 4 shows the relative error in the area estimate obtained with 100 evaluations of CLoVER. This figure also shows the evolution of the contour entropy.

## 4.2 Stability of tubular reactor

We use CLoVER to locate the stability boundary of a nonadiabatic tubular reactor with a mixture of two chemical species. This problem is representative of the operation of industrial chemical reactors,

and has been the subject of several investigations, e.g. [23]. The reaction between the species releases heat, increasing the temperature of the mixture. In turn, higher temperature leads to a nonlinear increase in the reaction rate. These effects, combined with heat diffusion and convection, result in complex dynamical behavior that can lead to self-excited instabilities. We use the dynamical model described in Refs. [24, 25]. This model undergoes a Höpf bifurcation, when the response of the system transitions from decaying oscillations to limit cycle oscillations. This transition is controlled by the Damköhler number $D$, and here we consider variations in the range $D \in [0.16, 0.17]$ (the bifurcation occurs at the critical Damköhler number $D_{cr} = 0.165$). To characterize the bifurcation, we measure the temperature at the end of the tubular reactor ($\theta$), and introduce the following indicator of stability.

$$g(D) = \begin{cases} \alpha(D), & \text{for decaying oscillations,} \\ (\gamma r(D))^2, & \text{for limit cycle oscillations.} \end{cases}$$

$\alpha$ is the growth rate, estimated by fitting the temperature in the last two cycles of oscillation to the approximation $\theta \approx \theta_0 + \bar{\theta}e^{\alpha t}$, where $t$ denotes time. Furthermore, $r$ is the amplitude of limit cycle oscillations, and $\gamma = 25$ is a parameter that controls the intensity of the chemical reaction.

Our goal is to locate the critical Damköhler number using two numerical models of the tubular reactor dynamics. The first model results from a centered finite-difference discretization of the governing equations and boundary conditions, and corresponds to IS0. The second model is a reduced-order model based on the combination of proper orthogonal decomposition and the discrete empirical interpolation method, and corresponds to IS1. Both models are described in details by Zhou [24].

Figure 5 shows the samples selected by CLoVER, and the uncertainty predicted by the GP surrogate at several iterations. The algorithm starts with two random evaluations of both models. This information is used to compute a MAP estimate of the hyperparameters of the GP surrogate, using the procedure recommended by Poloczek et al. [3][2] and to provide an initial estimate of the surrogate. In this example we use covariance kernels of the Matérn class [22] with $\nu = 5/2$, and zero mean functions.

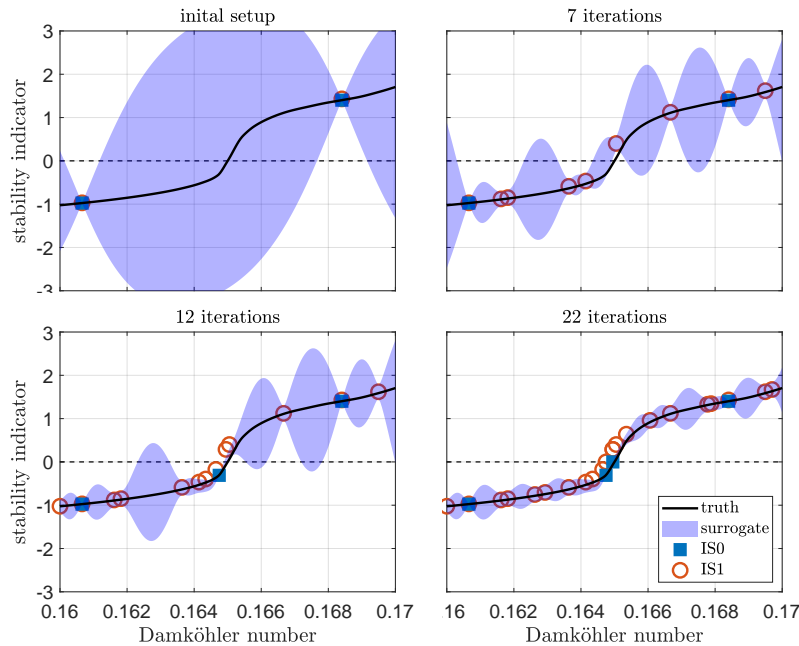

Figure 5: Locating the Höpf bifurcation of a tubular reactor (zero contour of stability indicator). Shaded area: $\pm 3\sigma$ around the mean of the GP surrogate. CLoVER locates the bifurcation after 22 iterations, using only 4 evaluations of IS0.

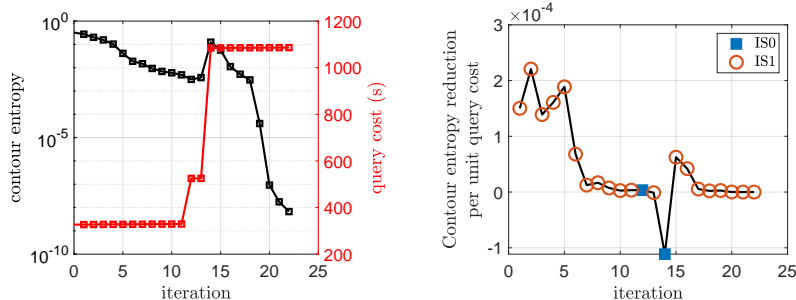

Figure 6: Left: Contour entropy and query cost during the iterations of the CLoVER algorithm. Right: Reduction in contour entropy per unit query cost at every iteration. CLoVER explores IS1 to decrease the uncertainty about the location of the bifurcation before using evaluations of expensive IS0.

After these two initial evaluations, CLoVER explores the parameter space using 11 evaluations of IS1. This behavior is expected, since the query cost of IS0 is 500-3000 times the query cost of IS1. Figure 6 shows the evolution of the contour entropy and query cost along the iterations. After an exploration phase, CLoVER starts exploiting near $D = 0.165$. Two evaluations of IS0, at iterations 12 and 14, allow CLoVER to gain confidence in predicting the critical Damköhler number at $D_{cr} = 0.165$. After eight additional evaluations of IS1, CLoVER determines that other bifurcations are not likely in the parameter range under consideration. CLoVER concludes after a total of 22 iterations, achieving $\mathscr{H} = 6 \times 10^{-9}$.

## 4.3 Comparison between CLoVER and existing algorithms for single information source

Here we compare the performance of CLoVER with a single information source to those of algorithms EGRA [15], Ranjan [16], TMSE [17], TIMSE [18], and SUR [18]. This comparison is based on locating the contour $g = 80$ of the two-dimensional Branin-Hoo function [26] within the domain $\mathcal{D} = [-5, 10] \times [0, 15]$. We discuss a similar comparison, based on a different problem, in the supplementary material.

The algorithms considered here are implemented in the R package `KrigInv` [19]. Our goal is to elucidate the effects of the distinct acquisition functions, and hence we execute `KrigInv` using the same GP prior and schemes for optimization and integration as the ones used in CLoVER. Namely, the GP prior is based on a constant mean function and a squared exponential covariance kernel, and the hyperparameters are computed using MLE. The integration over $\mathcal{D}$ is performed with the trapezoidal rule on a $50 \times 50$ uniform grid, and the optimization set $\mathcal{A}$ is composed of a $30 \times 30$ uniform grid. All algorithms start with the same set of 12 random evaluations of $g$, and we repeat the computations 100 times using different random sets of evaluations for initialization.

We compare performance by computing the area of the set $S = \{\boldsymbol{x} \in \mathcal{D} \mid g(\boldsymbol{x}) > 80\}$. We compute the area using Monte Carlo integration with $10^6$ samples, and compare the results to a reference value computed by averaging 20 Monte Carlo estimates based on evaluations of $g$: area$(S) = 57.8137$. Figure 7 compares the relative error in the area estimate computed with the different algorithms. All algorithms perform similarly, with CLoVER achieving a smaller error on average.

### Acknowledgments

This work was supported in part by the U.S. Air Force Center of Excellence on Multi-Fidelity Modeling of Rocket Combustor Dynamics, Award FA9550-17-1-0195, and by the AFOSR MURI on managing multiple information sources of multi-physics systems, Awards FA9550-15-1-0038 and FA9550-18-1-0023.

## Footnotes

[1]The statistical model used in the present algorithm is the same introduced in [3], and we attempt to use a notation as similar as possible to this reference for the sake of consistency.

[2]For the length scales of the covariance kernels, Poloczek et al. [3] recommend using normal distribution priors with mean values given by the range of $\mathcal{D}$ in each coordinate direction. We found this heuristics to be only appropriate for functions that are very smooth over $\mathcal{D}$. In the present example we adopt $d_0 = 0.002$ and $d_1 = 0.0005$ as the mean values for the length scales of $\Sigma_0$ and $\Sigma_1$, respectively.

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
