[Supplementary Material]

# Contour location via entropy reduction leveraging multiple information sources

**Alexandre N. Marques**    **Remi R. Lam**    **Karen E. Willcox**

*Supplementary material*

## A    Additional comparison between CLoVER and existing algorithms for single information source

Here we compare the performance of CLoVER with a single information source to those of the algorithms EGRA [15], Ranjan [16], TMSE [17], TIMSE [18], and SUR [18], similarly to case discussed in Sect. 4.3. In this investigation, we solve the problem described in example 1 of Ref. [15] (multimodal function). Consider the random variable $\boldsymbol{x} \sim \mathcal{N}(\boldsymbol{\mu}_x, \boldsymbol{\Sigma}_x)$, where

$$\boldsymbol{\mu}_x = \begin{Bmatrix} 1.5 \\ 2.5 \end{Bmatrix}, \qquad\qquad \boldsymbol{\Sigma}_x = \begin{bmatrix} 1 & 0 \\ 0 & 1 \end{bmatrix},$$

in the domain $\mathcal{D} = [-4, 7] \times [-3, 8]$. The goal is to estimate the probability $p_f = P(g(\boldsymbol{x}) > 0)$, where

$$g(\boldsymbol{x}) = \frac{(x_1^2 + 4)(x_2 - 1)}{20} - \sin\left(\frac{5x_1}{2}\right) - 2.$$

We estimate the probability $p_f$ by first locating the zero contour of $g$ and then computing a Monte Carlo integration based on the surrogate model:

$$p_f \approx \frac{1}{N} \sum_{i=1}^{N} \mathbb{I}_f(\boldsymbol{x}_i), \quad \boldsymbol{x} \sim \mathcal{N}(\boldsymbol{\mu}_x, \boldsymbol{\Sigma}_x).$$

where

$$\mathbb{I}_f(\boldsymbol{x}_i) = \begin{cases} 1, & \mu(0, \boldsymbol{x}_i) > 0, \\ 0, & \text{otherwise}, \end{cases}$$

$\mu(0, \boldsymbol{x}_i)$ denotes the mean of $f(0, \boldsymbol{x}_i)$, and $N = 10^6$ is the number of Monte Carlo samples. We assess the accuracy of the estimates by comparing them to a reference value computed by averaging 20 Monte Carlo estimates based on evaluations of $g$: $p_f = 0.03133$.

The R package `KigInv` [19] provides implementations of the algorithms listed above. As in Sect. 4.3, we execute `KrigInv` using the same GP prior and schemes for optimization and integration as the ones used in CLoVER. Namely, the GP prior is based on a constant mean function and a squared exponential covariance kernel, and the hyperparameters are computed using MLE. The integration over $\mathcal{D}$ is performed with the trapezoidal rule on a $50 \times 50$ uniform grid, and the optimization set $\mathcal{A}$ is composed of a $30 \times 30$ uniform grid. All algorithms start with the same set of 10 random evaluations of $g$, and stop when the acquisition function reaches a value of $10^{-8}$ or after 50 function evaluations, whichever occurs first. We repeat the computations 100 times using different random sets of evaluations for initialization.

Figure S.1 shows the relative error of the estimates of $p_f$. We observe that on average CLoVER results in a faster error decay, and converges to lower error level. The median number of function evaluations are the following. CLoVER: 38, EGRA: 42, Ranjan: 42, TMSE: 41, TIMSE: 50, SUR: 33.

We also evaluate the algorithms by computing the area of the subdomain $S = \{\boldsymbol{x} \in \mathcal{D} \mid g(\boldsymbol{x}) > 0\}$ (shaded area in Figure 3). We estimate the area using Monte Carlo integration with $10^6$ samples in the region $[-4, 7] \times [1.4, 8]$, and compare the results to a reference value computed by averaging 20 Monte Carlo estimates based on evaluations of $g$: $\text{area}(S) = 36.5541$. Figure S.2 shows the relative error in the estimates of the area of the set $S$. CLoVER also presents a faster decay of the error of the area estimate.

Figure S.1: Relative error in the estimate of the probability $p_f$ (median, 25th, and 75th percentiles). Left: comparison between CLoVER and greedy algorithms EGRA, Ranjan, and TMSE. Right: comparison between CLoVER and one-step look ahead algorithms TIMSE and SUR.

Figure S.2: Relative error in the estimate of the area of set $S$ (median, 25th, and 75th percentiles). Left: comparison between CLoVER and greedy algorithms EGRA, Ranjan, and TMSE. Right: comparison between CLoVER and one-step look ahead algorithms TIMSE and SUR.

# B   Trade-off between exploration and exploitation

As disussed in Sect. 3, the concept of contour entropy uses the parameter $\epsilon$ as a tolerance in the definition of the zero contour. This parameter also provides a control over the trade-off between exploration (sampling in regions where uncertainty is large) and exploitation (sampling in regions of relatively low uncertainty, but likely close to the zero contour). An algorithm that favors exploration may be ineffecient because it evaluates many samples in regions far from the zero contour, whereas an algorithm that favors exploitation may fail to identify disjoint parts of the contour because it concentrates samples on a small region of the domain. In general, larger values of $\epsilon$ result in more exploration than exploitation, and vice-versa.

We find that making $\epsilon$ proportional to the standard deviation of the surrogate model,

$$\epsilon(\boldsymbol{x}) = c_\epsilon \sigma(\boldsymbol{x}),$$

provides a good balance between exploration and exploitation. To determine the constant of proportionality $c_\epsilon$ we repeat the experiment described in Sect. 4.3 with $c_\epsilon \in \{1, 2, 3\}$. We measure the accuracy in the prediction of the zero contour by computing the area of the excursion set $S = \{\boldsymbol{x} \in [-5, 10] \times [0, 15] \mid g(\boldsymbol{x}) > 80\}$, where $g$ denotes the two-dimensional Branin-Hoo function [26]. Figure S.3 shows the convergence in the relative error in the estimates of $S$ computed with different values of $c_\epsilon$.

We observe that in all cases CLoVER identified the zero contour, although with varying levels of accuracy. As expected, $c_\epsilon = 1$ leads to a more exploitative algorithm that on average converges

Figure S.3: Influence of $\epsilon$ in the convergence of CLoVER. The plot shows the relative error (median, 25th and 75th percentiles) in the area of the set $S$.

faster to the zero contour. Choosing $c_\epsilon = 2$ does not affect the convergence rate significantly, but leads to a slightly larger error in the area estimate. Finally, setting $c_\epsilon = 3$ considerably degrades the performance of the algorithm.

Although we do not observe adverse effects of choosing $c_\epsilon = 1$ in this particular example, we also do not observe significant improvements with respect to $c_\epsilon = 2$. For this reason, we favor choosing $c_\epsilon = 2$ to avoid an excessively exploitative algorithm. We find this heuristic to work well in general problems.

## C Computational cost

The computational cost of CLoVER is comparable to that of other algorithms with one-step look ahead acquisition functions (e.g., TIMSE and SUR [18]). Greedy acquisition functions are cheaper to evaluate, but offer no natural form of selecting information sources when more than one is available. In addition, Chevalier et al. [19] report that one-step look ahead strategies are more efficient in selecting samples because they take into account global effects of new observations, resulting in a lower number of function evaluations for comparable accuracy. Most importantly, the multi-information source setting considered in this paper is relevant when the highest fidelity information source is expensive. In this scenario, the cost of selecting new samples ($\sim 4s$ for a two-dimensional problem) is normally small in comparison to function evaluations.

The computational cost of CLoVER is dominated by evaluating the variance $\bar{\sigma}^2$ within the integral of Eq. (10) (see Sect. 3.3). In general, the cost of evaluating the variance of a GP surrogate after $n$ observations is $\mathcal{O}(n^3)$. (The cost can be reduced to $\mathcal{O}(n \log^2 n)$ for specific covariance functions, and large values of $n$). Therefore, the total computations cost scales as $\mathcal{O}(n_a n_i n^3)$, where $n_a$ denotes the number of points in the optimization set $\mathcal{A}$, and $n_i$ denotes the number of points used to evaluate the integral of Eq. (10).