[Reviews · NeurIPS 2018]

Reviewer 1



This paper introduces an active learning strategy for estimating some contour line of an expensive to evaluate function in the case where lower fidelity evaluations of the objective function can be performed at lower cost. The proposed worklow relies on a multi-information source Gaussian Process model following notably the approach of Poloczek et al. (on "Multi-information source optimization") presented at NIPS 2017. Here however the acquisition function substantially differs from the one used in the former reference, and focuses specifically on contour line estimation rather than optimization. The proposed acquisition function builds upon the entropy of discrete distributions indicating at each point whether the response is in a prescribed interval around the contour level, below the interval lower bound or above the interval upper bound. At any given point, the entropy of this distribution can be written as a sum of three terms written in closed form (up to the Gaussian CDF) as a function of the predictive GP mean and standard deviation. Then this entropy is integrated over the input space (up to a normalization by the domain volume) and the considered sampling criterion is obtained at any given candidate evaluation point and fidelity level as the ratio between the expected decrease in this integrated entropy if a corresponding evaluation is performed and the (known) cost of performing it. While the integrand is not calculated in closed form, a tight approximation is provided for which a closed form expression is presened in the paper. An algorithm is then devised that consists, from a starting design, in performing at every iteration a new evaluation at selected point and fidelity level so as to achieve optimal resource allocation in the sense of the proposed criterion, until budget exhaustion. The contour line of the GP predictive mean at the final state is finally returned. The algorithm is tested and illustrated on two examples, one 2-dimensional analytical one and one 1-dimensional test case stemming from a chemical engineering problem. As motivated at the beginning of the paper, contour line estimation is an important topic arising in a number of applications and methodological contexts. Also, integrating several information sources withing a GP model in order to study expensive to evaluate functions under a smart resource allocation scheme is a timely and useful approach with promising benefits in a variety of settings. Adressing contour line estimation using a multi-information source GP and a dedicated acquisition function thus appears as a legitimate and worthwhile development. From that perspective, I very much welcome the present contribution. This being said, the paper also has some weaknesses, and while I recommend acceptance, I think that it deserves to be substantially improved before publication. One of the main concerns is that part of the relevant literature has been ignored, and also importantly that the proposed approach has not really been extensively compared to potential competitors (that might need to be adapted to the multi-source framework; not e also that single-fidelity experiments could be run in order to better understand how the proposed acquisition function compares to others from the literature). Another main concern in connection with the previous one is that the presented examples remain relatively simple, one testcase being an analytical function and the other one a one-dimensional mototonic function. While I am not necessarily requesting a gigantic benchmark or a list of complicated high-dimensional real-world test cases, the paper would significantly benefit from a more informative application section. Ideally, the two aspects of improving the representativity of numerical test cases and of better benchmarking against competitor strategies could be combined. As of missing approaches from the literature, some entry points follow: * Adaptive Designs of Experiments for Accurate Approximation of a Target Region (Picheny et al. 2010) http://mechanicaldesign.asmedigitalcollection.asme.org/article.aspx?articleid=1450081 *Fast kriging-based stepwise uncertainty reduction with application to the identification of an excursion set (Chevalier et al. 2014a) http://amstat.tandfonline.com/doi/full/10.1080/00401706.2013.860918 * NB: approaches from the two articles above and more (some cited in the submitted paper but not benchmarked against) are coded for instance in the R package "KrigInv". The following article gives an overview of some of the package's functionalities: KrigInv: An efficient and user-friendly implementation of batch-sequential inversion strategies based on kriging (Chevalier et al. 2014b) * In the following paper, an entropy-based approach (but not in the same fashion as the one proposed in the submitted paper) is used, in a closely related reliability framework: Gaussian process surrogates for failure detection: A Bayesian experimental design approach (Wang et al. 2016) https://www.sciencedirect.com/science/article/pii/S002199911600125X * For an overall discussion on estimating and quantifying uncertainty on sets under GP priors (with an example in contour line estimation), see Quantifying Uncertainties on Excursion Sets Under a Gaussian Random Field Prior (Azzimonti et al. 2016) https://epubs.siam.org/doi/abs/10.1137/141000749 NB: the presented approaches to quantify uncertainties on sets under GP priors could also be useful here to return a more complete output (than just the contour line of the predictive GP mean) in the CLoVER algorithm. * Coming to the multi-fidelity framework and sequential design for learning quantities (e.g. probabilities of threshold exceedance) of interest, see notably Assessing Fire Safety using Complex Numerical Models with a Bayesian Multi-fidelity Approach (Stroh et al. 2017) https://www.sciencedirect.com/science/article/pii/S0379711217301297?via%3Dihub Some further points * Somehow confusing to read "relatively inexpensive" in the abstract and then "expensive to evaluate" in the first line of the introduction! * L 55 "A contour" should be "A contour line"? * L88: what does "f(l,x) being the normal distribution..." mean? * It would be nice to have a point-by-point derivation of equation (10) in the supplementary material (that would among others help readers including referees proofchecking the calculation). * About the integral appearing in the criterion, some more detail on how its computation is dealt with could be worth. ##### added after the rebuttal I updated my overall grade from 7 to 8 as I found the response to the point and it made me confident that the final paper would be improved (by suitably accounting for my remarks and those of the other referees) upon acceptance. Let me add a comment about related work by Rémi Stroh. The authors are right, the Fire Safety paper is of relevance but does actually not address the design of GP-based multi-fidelity acquisition functions (in the BO fashion). However, this point has been further developed by Stroh et al; see "Sequential design of experiments to estimate a probability of exceeding a threshold in a multi-fidelity stochastic simulator" in conference contributions listed in http://www.l2s.centralesupelec.fr/en/perso/remi.stroh/publications

Reviewer 2



This paper presents a methodology to accurately estimate a level set of an expensive to evaluate function, with several levels of fidelity/cost available. A new acquisition function is introduced, defining an entropy for the contour location. I am not aware of many works with multi-fidelity (pertinent for Machine Learning) in this context, but there is a closely related one that should be discussed: Bogunovic, Ilija, et al. "Truncated variance reduction: A unified approach to Bayesian optimization and level-set estimation." Advances in Neural Information Processing Systems. 2016. Also, an acquisition function targeting the contour itself is proposed in: Chevalier, C., Ginsbourger, D., Bect, J., & Molchanov, I. (2013). Estimating and quantifying uncertainties on level sets using the Vorob’ev expectation and deviation with Gaussian process models. In mODa 10–Advances in Model-Oriented Design and Analysis (pp. 35-43). Springer, Heidelberg. For closed form expressions of [15], see e.g.,: Chevalier, C., Bect, J., Ginsbourger, D., Vazquez, E., Picheny, V., & Richet, Y. (2014). Fast parallel kriging-based stepwise uncertainty reduction with application to the identification of an excursion set. Technometrics, 56(4), 455-465. It would be nice to compare the relative costs of those approaches, with their corresponding approximations. The contour entropy depends on the parameter epsilon, with a proposed default of 2*sigma(x). Could you show that the performance is not too sensitive to this choice. In particular, a fixed value should still work since the Gaussian predictive distribution shrinks with new observations anyway. The experiments are convincing, but I am curious about what is the most important: leveraging several levels of fidelity or the acquisition function? Especially, the contour entropy is quite computationally expensive, while simple criteria (e.g., from [13] and [14]) are cheap. ## Additions after rebuttal The authors’ response to the various points (shared among reviewers) is complete and suggests that the final version will be improved. I thus increased my evaluation score (7 to 8).

Reviewer 3



This paper presents the concept of contour entropy which helps locating the contour of a function using multiple information sources under limited budget. Main strengths: - Definition of contour entropy - Derivation of acquisition function for multiple information sources - Interesting simulation and real world experiment Main weaknesses: - Missing references to several areas of Machine Learning (see below) - No comparison with baseline approaches (there is one in the appendix, but it does not seem to be showing significant improvement) It looks like the authors have a good grasp of the contour location literature, but there are several areas of ML that are related to the overall problem studied here, which should be referenced. The acquisition function derived in eqn 6 and 7 are basically a form of expected utility, and it has been used in a similar context by Melville et al. An Expected Utility Approach to Active Feature-value Acquisition http://www.prem-melville.com/publications/afa-icdm-05.pdf (Please also see their follow up work) Another related area in the context of learning from multiple weak, and possibly noisy approximations of a function is ensamble methods. I would again refer to a thesis by Prem Melville, Creating Diverse Ensemble Classifiers to Reduce Supervision. On the experimental front, the simulation experiment is interesting, as is the experiment from chemical engineering domain. However, it's not clear what you're comparing your proposed approach to. I would recommend your baseline comparison from appendix to the main paper, although those results do not seem to be significantly better than the baseline. Question for authors: line 138: for the evaluation of the entropy, you mention that it is deterministic from equations (4) and (5). Are these always tractable for complex functions?